# The Diversity and Zoonotic Potential of *Staphylococcus pseudintermedius* in Humans and Pet Dogs in Trinidad and Tobago

**DOI:** 10.3390/antibiotics12081266

**Published:** 2023-07-31

**Authors:** Sharianne Suepaul, Marc Stegger, Filip Boyen, Karla Georges, Patrick Butaye

**Affiliations:** 1Department of Pathobiology, School of Veterinary Medicine, St. George’s University, True Blue, St. George’s FZ818, Grenada; 2Department of Basic Veterinary Sciences, School of Veterinary Medicine, Faculty of Medical Sciences, The University of the West Indies, St. Augustine 685509, Trinidad and Tobago; karla.georges@sta.uwi.edu; 3Department of Bacteria, Parasites, and Fungi, Statens Serum Institut, 2300 Copenhagen, Denmark; mtg@ssi.dk; 4Antimicrobial Resistance and Infectious Diseases Laboratory, Harry Butler Institute, Murdoch University, Perth, WA 6150, Australia; 5Department of Pathobiology, Pharmacology and Wildlife Medicine, Faculty of Veterinary Medicine, Ghent University, 9820 Merelbeke, Belgium; filip.boyen@ugent.be (F.B.); or pabutaye@cityu.edu.hk (P.B.); 6Department of Infectious Diseases and Public Health, Jockey Club College of Veterinary Medicine and Life Sciences, City University of Hong Kong, Kowloon, Hong Kong; 7School of Veterinary Medicine, Ross University, Basseterre P.O. Box 334, Saint Kitts and Nevis

**Keywords:** *Staphylococcus pseudintermedius*, methicillin-resistant, diversity, virulence, dog, human, comparative genomics

## Abstract

*Staphylococcus pseudintermedius* is an opportunistic pathogen that is frequently isolated from canines. It is of escalating interest because of its increasing antimicrobial resistance and zoonotic potential. Although many published articles are available that describe isolates obtained from diseased dogs and humans, this study focused on isolates obtained from healthy dogs and their owners who presented at clinics for routine veterinary care and utilized whole genome sequencing-based analyses for strain comparisons. A total of 25 humans and 27 canines were sampled at multiple sites, yielding 47 and 45 isolates, respectively. Whole genome sequence analysis was performed. We detected mostly new sequence types (STs) and a high diversity. Strains carried few antimicrobial resistance genes and plasmids, albeit three MRSP strains were found that belonged to two internationally distributed STs. The virulence content did not provide insights toward a tendency to colonization of humans but supported that there may be differences in the surface proteins between carrier strains and those causing pyoderma. We identified 13 cases in which humans were infected with strains from the dog they owned.

## 1. Introduction

Staphylococci are opportunistic organisms that can cause infections in humans and animals [1]. This genus is mainly investigated for its molecular epidemiology, virulence [2,3], and antimicrobial resistance [4,5]. The current study focused on *Staphylococcus pseudintermedius,* which commonly causes pyogenic soft tissue infections, otitis externa, sinusitis, osteomyelitis, endocarditis, and post-operative abscesses [6,7,8]. *S. pseudintermedius* regularly colonizes dogs and cats at multiple sites such as the skin and mucous membranes [9,10] but may also be isolated from horses [11] and humans [12].

Dogs are of particular interest since up to 90% of dogs have been shown to carry this bacterium [13]. Additionally, different strains of *S. pseudintermedius* can inhabit different sites of a dog at the same time, which has vast implications for diagnostic testing and the determination of antimicrobial sensitivity [14]. This organism can be zoonotic because it can be transferred from pet dogs to their human owners, and there is the potential for *S. pseudintermedius* to cause infections in humans, particularly in immunocompromised individuals [15]. Although carriage of *S. pseudintermedius* may be short-term and sporadic in healthy humans, dogs can be carriers for extended periods of time [13].

Antimicrobial resistance in *S. pseudintermedius* is of growing concern, especially the methicillin-resistant *S. pseudintermedius* (MRSP) strains [16,17,18,19]. Like methicillin-resistant *Staphylococcus aureus* (MRSA), MRSP has evolved via the horizontal transfer, acquisition, and insertion of the staphylococcal cassette chromosome (SCC), which carries a *mec* gene (*mecA* or *C*) as well as other genes encoding virulence characteristics, other resistance genes, and metal resistances [20]. The *mecA* gene encodes an alternative penicillin-binding protein (PBP2a) that has a low affinity for β-lactam antibiotics [21], leading to resistance to these commonly used antibiotics. MRSP strains have also been associated with resistance to multiple classes of antimicrobials and hence are termed multidrug-resistant (MDR). The occurrence of these antimicrobial-resistant strains is unsettling because it limits the options for therapeutic management of infections and has a grave impact on morbidity and mortality [6].

Recently, there has been increasing evidence of the role of MRSP in causing infections in canines as well as humans [22], including the transmission of *Staphylococcus pseudintermedius* between humans [23,24]. This is mainly due to the advancement in the techniques used to identify the *Staphylococcus* species, which include matrix-assisted laser desorption/ionization–time of flight (MALDI-TOF) mass spectrometry and the combination of molecular subtyping and sequencing techniques that aid in increasing the accuracy of identifying *S. pseudintermedius* infections [13]. As a result, there is a rapidly growing pool of genomic data available regarding *S. pseudintermedius*. Most of the data involve MRSP, which displays a rather a clonal population structure [4]. Although there are many different sequence types, a geographical pattern of distribution was observed by Perreten et al. in 2010 [25], when ST71 was the major clone observed in Europe and ST68 was the major clone observed in the USA. Recent studies published in 2022 indicated that ST71 is currently the major MRSP clone observed globally [4,26,27].

There is an increasing interest in identifying and comparing the various virulence factors and colonization capacities possessed by the various sequence types of both MRSP and methicillin-susceptible *S. pseudintermedius* (MSSP) to determine if there are any associations with in vivo virulence. Thus far, there are indications that the surface proteins (*spsD/F/P/Q*) that are involved in colonization might be predominant in those isolates that cause pyoderma in dogs [28], indicating that these isolates may exhibit a higher pathogenicity. The genes encoding for *spsL* and *spsD* have been associated with host specificity; however, further research is required for confirmation because these genes show a high sequence variation [29].

The objectives of this study were to elucidate the clonal types of *S. pseudintermedius* present in Trinidad and Tobago and to determine the zoonotic potential and specific characteristics of the accessory genome of strains isolated from healthy dogs and their owners using genome sequencing.

## 2. Results

### 2.1. Strains

Seventy-two strains were included in this study. They originated from 27 apparently healthy dogs and 25 owners of these dogs, representing 25 human–dog pairs and 2 dogs for which their owner was negative. Forty-five strains originated from dogs, and twenty-seven were from humans. Of the dog strains, 17 were isolated from the nose, 18 were from the mouth, and 10 were from the skin of the abdomen. Of the human strains, 8 were isolated from the nose, 5 were from the mouth, and 14 were from the hands (Table 1).

### 2.2. MLST

Only seven previously identified STs (ST1709, ST1097, ST373, ST758, ST71, ST45, and ST192) representing 12 strains were found amongst the 72 strains. All other isolates presented novel sequence types, as shown in Figure 1. One strain could not be typed due to absence of the *pta* gene. An eBURST analysis demonstrated that the strains were singletons.

### 2.3. Phylogeny

In all cases, not a single dog–human combination contained exactly the same strain, with all showing at least one SNP difference (Appendix A). In 13 cases, the SNP differences were between 1 and 12, indicating that both isolates represented the same clone or strain. A second group of two cases could be identified with the number of SNPs between 12 and 50, and finally a third group with more than 50 SNPs included nine cases. In the latter group, no transfer of the strain could be demonstrated, while in the second group, it was unclear whether these strains represented the same clone and transmission had occurred. However, multiple strains colonizing the same dog (as well as multiple strains in a human in a few cases) were observed. In one dog, two identical strains were isolated from different sampling sites (Figure 1).

We isolated *S. pseudintermedius* at multiple sites on dogs (nose, mouth, and ventral abdomen) and obtained several isolates from the same dog. We obtained multiple strains (two to four strains) from 15 dogs; in 5 of the cases the isolates represented the same strain (less than 12 SNPs), and in 10 cases, the isolates were different strains.

In contrast, although we sampled multiple anatomical sites in humans (nose, mouth and hands), we isolated significantly (chi-squared; *p* = 0.008) fewer strains from different sites from a single human than from dogs. We obtained two isolates from five people, and in four of those cases, the isolates represented different strains (more than 12 SNPs). In one single case, the same strain was isolated twice.

### 2.4. Antimicrobial Resistance (AMR) Genes

It is important to note that the isolates utilized in this study were not independent because we isolated them from owners and dogs in the same household as well as multiple clones from one dog. As such, there was a bias in the dataset for duplicates. Seventeen strains were fully susceptible, and a few strains carried multiple resistance genes. Forty-five out of seventy-two strains carried a *blaZ*, and three MRSPs were identified: two ST71s from a dog and his owner and one ST45 from the nose of a dog. The *mecA* gene in the ST71 strain was located in the SCC*mec* type III(3A), while the SCC*mec* element of the ST45 strain was non-typeable as commonly noted in CC45 strains [25]. A BLASTN analysis of the formerly described non-categorized ψSCCmec_57395_ revealed a 100% identity with this element; when using easyfig [30], the structure was shown to be the same. Thirty strains were *tet*(M)-positive, and two strains (the MRSP ST72) carried the *tet*(K) gene. Fifteen strains carried the *dfrG* gene. All other AMR genes were only carried by between one and six strains. Resistance to quaternary ammonium compounds mediated by the *qacG* gene was carried by ten strains (Figure 1).

### 2.5. Plasmids

Few strains carried plasmids, and even fewer carried more than one. A total of 17 strains carried one or more plasmid replicons. One strain carried two plasmids, and the MRSP strain ST45 carried four plasmids. The most common plasmid replicon found was the repUS43 replicon, which was carried by 15 strains. The second most common was the rep7a replicon, which was carried by four strains, of which two also carried the repUS43 replicon. The additional replicons found (repUS12 and rep13) were carried by the ST45 MRSP that also carried the other two replicons.

### 2.6. Virulence Genes

*S. pseudintermedius* virulence genes were found in all strains (Figure 1). Using the VFDBins, only a few enterotoxins (*sea*, *sec*, *selI*, *selK*, and *selq*) were found in a few strains. Using the *S. pseudintermedius* database to compare these strains, it was found that all strains contained the enterotoxin-encoding genes (*seaR* and *seaS*), all but three contained the *se-int*, and only few contained the *sec-canine*. The accessory gene regulator genes (*agrA* and *agrD*) were found in all strains, while the *agrB* gene was frequently absent. All four described types of *agrD* were detected. The most common type was type III with 22 strains that were positive. Types I and IV were found in 17 strains, and type II was found in 14 strains. Other virulence gene regulators present in all strains were the global regulator (*rot, srrA*, and *sigB*), and the *traP* and *sarA* genes were detected in all but one strain. All strains also contained genes encoding leukotoxins (lukF-I and lukS-I), proteases (*clpP* and *clpX*), the elastin binding protein gene (*ebpS*), β-hemolysin (*hlb*), and all but one surface protease (*htrA*). While all strains contained the exfoliative toxin gene (*siet*), all but one contained the *speta* gene, few contained the exfoliative toxin genes *exi* (10) and *expB* (9), and none the *expA* gene. Most of the strains had the capacity to form biofilms since they contained the *icaA/B/C/D* genes (63/72). All but two strains contained the nuclease gen *nucC,* and slightly more than half contained the coagulase gene *coa*. The sialidase encoding gene *nanB* was found in 13 strains. Several staphylococcal surface proteins were present, of which some were in all strains while others were in fewer strains. The *spsJ*, *spsO*, and *spsP* genes were not detected.

### 2.7. Prophages

All strains carried prophages, of which some were only incomplete phages. Due to the multitude of incomplete phages, it was not possible to make concise comparisons. However, similar-sized phages were found in the dog–owner pairs as well as similar sizes of incomplete phages. There was certainly a great variety of different sizes of prophages amongst the strains within the same ST and between the different STs.

## 3. Discussion

In contrast to most other studies that focused on MRSP, we sequenced a collection of colonizing strains that included the strains that also infected the owner of the dog. To our knowledge, there is only one study that used WGS and thus unambiguously showed transmission [23]. We studied apparently healthy animals that came to veterinary clinics for routine checkups or vaccinations.

### 3.1. Genetic Diversity

Most of the strains represented new STs and were singletons indicating a geographically specific population structure of *S. pseudintermedius* in Trinidad. New STs of *S. pseudintermedius* from dogs have been frequently reported in as yet underexplored regions [27,31,32], and these studies showed a specific geographical association of *S. pseudintermedius* sequence types. In contrast, MRSP isolates were more common STs with the European MRSP ST71, which was identified in one dog–human pair, and the MRSP ST45, which was only found in a dog. This study detected the most reported MRSP clones (ST71 and ST45 [33]), indicating a worldwide expansion of these clones that includes the Caribbean region, probably through tourism or animal trade. While the US is geographically closer, the MRSP ST68 clone, which is typically associated with the US [33], was not detected. Nevertheless, in Trinidad, MRSP does not seem to be highly prevalent because most strains are MSSP.

The MRSP ST71 was also found on an owner, while the MRSP ST45 strain was not found on an owner. There were no real indications that there was a difference in colonization capacity between the two detected MRSP clones because we did not detect any difference in the virulence genes apart from the *coa* and *spsI* genes, which were present in the ST45 strain but not in the ST71 strain. Intriguingly, the human ST71 strain, while being nearly identical (only one SNP difference) in all investigated accessory genes, lacked the *spsD* gene. Whether this was associated with an adaptation after colonization of the owner remains to be determined.

One strain could not be typed using MLST because the *pta* gene was absent. The *pta* gene encodes a phosphate acetyltransferase, and to the best of our knowledge, the absence of that gene has never been reported previously. Assuming that the gene was incompletely sequenced, we were not able to locate potential parts of the gene in the sequence. Upon observation of the cluster pattern of the strain (N2S2DN; Figure 1), it was observed that it was not clustering together with a specific sequence type. Thus, we assumed that this gene was absent in this strain and that this was a rare finding.

It has been shown previously using different methods that the genetic diversity of strains in a single individual dog can be considerable [34]. Also, in our study, this was evident, although there were also quite a number of dogs with the same clone present. This was less obvious in humans, although it should be noted that there were also fewer humans from whom we could obtain multiple isolates using our methodology. We did not find any indication that certain *S. pseudintermedius* strains colonize humans better than animals.

Transmission of *S. pseudintermedius* from dogs to humans has been studied on several occasions [29,35]. The transfer, which has been demonstrated at different degrees, is also dependent on the sampling methods used as well as whether the dog was experiencing pyoderma, making it difficult to estimate transmission rates in general. The same/similar strains have been found to colonize both the animal and the owner, though colonization with unrelated strains has also been demonstrated [24,36,37]. High diversity may account for the variation amongst strains colonizing dogs and humans. Indeed, it has been shown that there was a large diversity of strains on the same dog and that the detection of types differed over time [38]. As such, the absence of the strain that was found on the owner and dog may be merely a result of this high diversity of strains and the limited number of isolates investigated from each dog in this study. A study assessing the diversity of strains from the same dog would be of interest.

### 3.2. AMR

In this study, only 3 out of 72 strains were methicillin-resistant, and the other strains were broadly susceptible to most antibiotics. The resistance against antimicrobials in *S. pseudintermedius* is quite variable according to the studies performed, though MRSP strains tend to be increasingly prevalent, and those strains tend to be multidrug-resistant [39,40], as also seen in our study. The AMR genes found in these strains are the classical resistance genes often found in *S. pseudintermedius* [41,42].

### 3.3. Plasmids

Though the resistance genes detected are frequently associated with plasmids [42], few plasmids were present, and most of the strains were devoid of plasmids. The repUS43 replicon has been found in different Gram-positive bacteria worldwide (assessed through a BLAST search of the sequence and has been shown to be associated with plasmids carrying the antimicrobial resistance gene, although in our study, no such association could be found, which may have been caused by the fact that the assembly did not allow for a closed sequence. Typically, the MRSP strains carried the rep7a, and only one MSSP carried a plasmid with this replicon; this replicon also has been found in a multitude of Gram-positive bacteria (via BLAST search). The MRSP ST45 carried most plasmids with all four different replicons we detected.

### 3.4. Virulence

The *S. pseudintermedius* strains isolated in this study seemed to have a core virulence gene content similar to previously studied strains [43]. Four genes coding for surface proteins (*spsD*, *spsF*, *spsP,* and *spsQ*) involved in colonization by binding to the host’s extracellular matrix were previously shown to be present mainly in dog pyoderma isolates [28], although this was not absolute. In our collection of strains, we detected these genes to a lesser extent than most of the other surface proteins (23, 1, 0, and 3, respectively). It is striking that three of those surface proteins were nearly absent, indicating that those genes may indeed be of importance in pyoderma since our strains were from healthy dogs. This indicated that the strains used in this study were mainly the typical colonizing strains and that these strains can readily colonize humans.

Differences in virulence and phage content have been found between human and dog MRSP strains of a same ST, including an ST45 strain [44] similar to that in our study. On the other hand, in a larger collection of strains, no differences could be found [29]. Unfortunately, the owner of the dog carrying this ST45 strain did not carry the same strain. The ST71 strain of our study showed a difference with the human strain lacking the *spsD* gene, while in the study by Phumthanakor et al. (2021) [44], the *spsP* and *spsQ* genes were differently present in the dog and human strains. Nevertheless, these surface proteins are probably associated with pyoderma in dogs. It may thus well be that these proteins were lost since they are specific for adhesion to dog keratinocytes. Differences were also seen in the phages; however, since our sequences were Illumina-generated sequences, the comparison of the presence of phages was not possible. Though several differences were noted, they might have been artificial because they were partially located on different contigs. Long-read sequencing could solve this issue.

Other differences were noted between strains of the same ST, and those were mainly seen in different surface proteins (*spsB/D/I/L/Q*/*R*). Other genes that differed were the enterotoxin *se-int* and the nuclease *nunC*. Differences in the presence was also seen for several *sps* genes within a sequence type, which has been noted before [43]. The reason why these surface proteins differ so much between clonally related strains compared to other genes remains to be elucidated.

## 4. Materials and Methods

### 4.1. Isolates

The isolates were obtained from dogs and their owners and identified using MALDI-TOF in a previous study [45]. Sampling in this former study was done on dog–owner combinations with the aim to isolate all Staphylococcaceae and determine the species distribution and overlaps as well as the antimicrobial susceptibility of the isolates. The *S. pseudintermedius* isolates included in this study were selected from this collection based on the presence of an isolate in a dog and owner. From several dogs and owners, several isolates were obtained from different locations and subsequently subjected to genome sequencing. There were isolates included from two dogs for which the owner isolate was excluded because of contamination of the DNA sample that became evident after sequence analysis.

### 4.2. Whole Genome Sequencing and Sequence Analysis

Overnight cultures were grown in tryptic soy broth at 37 °C with 200 rpm shaking. Genomic DNA was extracted using the DNeasy Blood and Tissue kit (Qiagen). Library preparation was carried out using the Nextera XT kit and paired-end 2 × 250 nbp sequencing on a MiSeq, all following standard Illumina protocols (Illumina Inc., San Diego, CA, USA). All raw reads were deposited under bioproject PRJNA778212 (BioSamples: SAMN22933514 and SAMN31370566-SAMN31370626).

De novo assembly was conducted using a unicycler, quality was assessed with QUAST, and sequences were annotated using RASTk on the Patric server (https://www.patricbrc.org/ accessed on 6 August 2022). MLST was performed on the CGE server using ‘MLST’ (https://cge.cbs.dtu.dk/services/MLST/, accessed for this manuscript on 12 August 2022) [46]. Unknown MLST profiles and alleles were submitted to the ‘Public databases for molecular typing and microbial genome diversity’ (https://pubmlst.org/ accessed for this manuscript on 2 November 2022), and new allele numbers and sequence types were obtained. MLST profiles were compared using BURST on pubMLST.

To investigate the genetic relatedness of the isolates, we used CSI Phylogeny 1.4 on the CGE server (http://www.genomicepidemiology.org/ accessed for this manuscript on 20 August 2022) for SNP analysis [47] using the *S. pseudintermedius* reference genome ATCC 49051 (named 1-44876). The relatedness of the isolates was visualized using ITol [48].

The following analyses were performed with pipelines from the Center for Genomic Epidemiology (http://www.genomicepidemiology.org/ accessed for this manuscript on 25 August 20222): Kmer analysis to confirm the species identification (KmerFinder) [49], ResFinder v.3.0 for the detection of resistance genes [50], PlasmidFinder v2.0 for the detection of plasmid replicons [51], and SCC*mec*Finder for identification of the SCC*mec* type. Sequences not included in the SCC*mec* typing scheme were downloaded, and BLASTN was used for finding similar sequences. Structure of the pseudoSCC*mec* elements was compared using easyfig [30]. Phages were analyzed using Phaster (https://phaster.ca/ accessed for this manuscript on 27 October 2022) [52].

Virulence genes were determined with Abricate using vfdb [53] and a database specifically developed for the detection of *S. pseudintermedius* virulence genes (DB_SP) [43]. The specific database contained the sequences for *spsA-R*, *clpP*, *siet*, *speta*, *se-int*, *lukF-I*, *lukS-I*, *sec-canine*, *exi*, *expB*, *agrA-D*, *icaA-D*, *nanB*, *coa*, *clpX*, *saeR*, *saeS*, *htrA*, *nucC*, *hlb*, *sigB*, *srrA*, *sarA*, *rot*, *traP*, *expA*, and *epbS*. We determined the four different *agrD* types (GenBank accession nos. EU157356.1, EU157391.1, EU157400.1, and EU157402.1) using BLASTN on the Patric server [34].

Statistical analysis of differences between prevalence of characteristics between dogs and humans were assessed via the chi-squared test.

## 5. Conclusions

Less than half of the sampled dog–owner combinations showed that humans were colonized with *S. pseudintermedius* strains of the healthy dog they owned (defined as less than 12 SNPs). In some cases, other strains were found, although this may merely have reflected the high diversity of *S. pseudintermedius* strains that can be present within one dog. We identified mostly new STs and a high diversity of *S. pseudintermedius* strains in Trinidad. Strains carried few antimicrobial resistance genes and few plasmids, albeit three MRSP strains were found belonging to two internationally distributed STs. There were no indications that strains isolated from owners possessed specific virulence genes that could facilitate the colonization of humans.

## Figures and Tables

**Figure 1 antibiotics-12-01266-f001:**
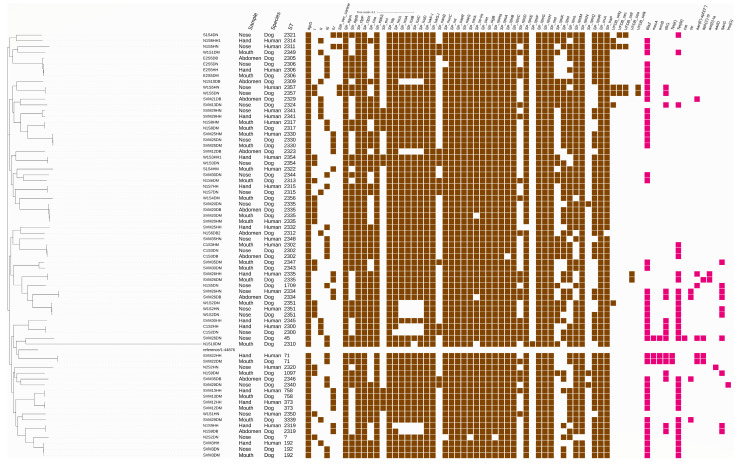
SNP-based phylogenetic tree and accessory genome of the analyzed *S. pseudintermedius* strains. New sequence types all had STs higher than 2300. Virulence genes are shown in brown, and resistance genes are all in purple. Samples are named according to their origin; the last letters signify whether the strain originated from a dog (D) or human (H) sampling site with N for nose, H for hand, B for abdomenal skin, and M for mouth. Strains with the same prefix were from the same dog–owner combination. The strain named reference 1-44876 was the *S. pseudintermedius* reference genome ATCC 49051.

**Table 1 antibiotics-12-01266-t001:** Origin of the strains investigated.

Animal	Number of Isolates	Isolation Site		
		Nose (n)	Mouth (n)	Abdomen (Dog) Hand (Human) (n)
Dog (n = 27)	45	17	18	10
Human (n = 25)	27	8	5	14

## Data Availability

The raw sequence data are available from the NCBI under BioProject PRJNA892162, BioSample SAMN22933514, and SRA numbers SRR21977355 to SRR21977426.

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
