# Peer review of "The Diversity and Zoonotic Potential of Staphylococcus pseudintermedius in Humans and Pet Dogs in Trinidad and Tobago"

_antibiotics, 2023, doi:10.3390/antibiotics12081266_

Round 1

Reviewer 1 Report

The study investigates Staphylococcus pseudintermedius, a pathogen commonly found in dogs, focusing on isolates obtained from healthy dogs and their owners. Whole genome sequencing was used to analyze the strains. The findings show a high diversity of mostly new sequence types (STs) with limited antimicrobial resistance genes and plasmids. However, a few strains with methicillin resistance were detected. The study did not find a clear pattern of strain colonization in humans based on virulence content but observed differences in surface proteins between carrier strains and those causing skin infections. Notably, there were instances where humans were infected with strains from their own dogs.

Overall, the manuscript provides valuable data to enhance the understanding of the diversity and zoonotic potential of Staphylococcus pseudintermediu. To improve readability and clarity, the following suggestions are provided:

Below are more specific comments:

Figure 1: The legend lacks detailed contents and could be improved (see related comment below).

Line 104: Consider color-coding the 12 previously identified STs strains in Figure 1 to help readers clearly observe the distribution of STs among the SNP-based phylogeny tree. From what I can see, most of the new STs belong to a separate tree clade, distinct from the 'old STs.' This observation may support the author's discussion regarding a geographically specific population structure of S. pseudintermedius in Trinidad.

Line 113: The current Figure 1 does not clearly depict the 'dog-human combination.' Please update the 'Species' column in the figure with 'Species-case tag.' For example, 'Dog-1' and 'Human-1' would indicate the human-dog couple from case 1.

Line 114-117: It is challenging to discern the SNP difference between each dog-human case in Figure 1 alone. Please include a table with columns for 'Strain ID,' 'Sample site,' 'Species,' 'Case ID,' and 'SNP difference (level 1, 2, 3).'

Line 128-129: Significantly fewer strains than what? It seems like "we isolated significantly fewer strains from different sites from a single person than from a single dog" (Chi-squared; p=0.008).

Line 149: I cannot find the related plasmid analysis result described in Figure 1. Please add the related result.

Line 205: “MRSP ST45 strain”.

Line 313-314: The article provides vague information about the construction method of the evolutionary tree. It does not mention the reference genome used, nor does it specifically explain the method of obtaining the number of SNP differences mentioned multiple times in the article. Please add these details to provide clarity.

Author Response

Reviewer 1

The study investigates Staphylococcus pseudintermedius, a pathogen commonly found in dogs, focusing on isolates obtained from healthy dogs and their owners. Whole genome sequencing was used to analyze the strains. The findings show a high diversity of mostly new sequence types (STs) with limited antimicrobial resistance genes and plasmids. However, a few strains with methicillin resistance were detected. The study did not find a clear pattern of strain colonization in humans based on virulence content but observed differences in surface proteins between carrier strains and those causing skin infections. Notably, there were instances where humans were infected with strains from their own dogs.

Overall, the manuscript provides valuable data to enhance the understanding of the diversity and zoonotic potential of Staphylococcus pseudintermedius. To improve readability and clarity, the following suggestions are provided:

We thank the reviewer for this concise assessment

Below are more specific comments:

Figure 1: The legend lacks detailed contents and could be improved (see related comment below).

We have adapted the legend of the figure.

Line 104: Consider color-coding the 12 previously identified STs strains in Figure 1 to help readers clearly observe the distribution of STs among the SNP-based phylogeny tree. From what I can see, most of the new STs belong to a separate tree clade, distinct from the 'old STs.' This observation may support the author's discussion regarding a geographically specific population structure of S. pseudintermedius in Trinidad.

We have included text on the new STs in the legend of the figure as they are all the STs that are higher than 2300. According to us, this will make it clearer that working with colors. Indeed there is a large clade of the new STs. Some however cluster with other STs. They are however singletons in eburst analysis (data that were not shown as not really informative since all were singletons)

Line 113: The current Figure 1 does not clearly depict the 'dog-human combination.' Please update the 'Species' column in the figure with 'Species-case tag.' For example, 'Dog-1' and 'Human-1' would indicate the human-dog couple from case 1.

This information is in the sample, we now explained that in the legend. We hope this is clear enough for the reader.

Line 114-117: It is challenging to discern the SNP difference between each dog-human case in Figure 1 alone. Please include a table with columns for 'Strain ID,' 'Sample site,' 'Species,' 'Case ID,' and 'SNP difference (level 1, 2, 3).'

Indeed, it is not possible in the figure, that is why we made 3 groups in the text. We indeed have an excel table with differences and the couples (dog-human) in highlight, though it is a very large table which does not really fit in the manuscript. We have added this table to the supplementary material (Table 1. SNP matrix).

Line 128-129: Significantly fewer strains than what? It seems like "we isolated significantly fewer strains from different sites from a single person than from a single dog" (Chi-squared; p=0.008).

This has been clarified in the text Lines 135-139

Line 149: I cannot find the related plasmid analysis result described in Figure 1. Please add the related result.

Indeed the plasmids are not in figure 1. It was however originally the plan, though upon seeing the figure, it was too large and made it unclear. We forgot however to delete this from the text. Is now deleted, Line 164

Line 205: “MRSP ST45 strain”.

Has been adapted now line 213

Line 313-314: The article provides vague information about the construction method of the evolutionary tree. It does not mention the reference genome used, nor does it specifically explain the method of obtaining the number of SNP differences mentioned multiple times in the article. Please add these details to provide clarity.

We have adapted and included the reference strain. As for the method CSI Phylogeny uses, it is not relevant to the manuscript. This is a copy paste of the method as it was published and included in the referenced website.

Reads were mapped to reference sequences using BWA v. 0.7.2 [20]. The depth at each mapped position was calculated using genomeCoverageBed, which is part of BEDTools v. 2.16.2 [21]. Single nucleotide polymorphisms (SNPs) were called using mpileup part of SAMTools v. 0.1.18 [22]. SNPs were filtered out if the depth at the SNP position was not at least 10x or at least 10% of the average depth for the particular genome mapping. The reason for applying a relative depth filter is to set different thresholds for sequencing runs that yield very different amounts of output data (total bases sequenced). SNPs were filtered out if the mapping quality was below 25 or the SNP quality was below 30. The quality scores were calculated by BWA and SAMTools, respectively. The scores are phred-based but can be converted to probabilistic scores, with the formula 10(Q/10), where Q is the respective quality score. The probabilistic scores will represent the probability of a wrong alignment or an incorrect SNP call, respectively. In each mapping, SNPs were filtered out if they were called within the vicinity of 10 bp of another SNP (pruning). A Z-score was calculated for each SNP as described above for NDtree.

Reviewer 2 Report

The study is interesting but they do not show results of the sampling sites between individuals and animals or between individuals, the statistics are poor and from all the samplings that were carried out they could present much more valuable information for comparisons between species.

They do not mention why they established the total number of samples, nor the power of the test.

The conclusion could be a little broader if they mentioned the differences between sampling sites and owner-dog relationships.

The study is interesting but they do not show results of the sampling sites between individuals and animals or between individuals, the statistics are poor and from all the samplings that were carried out they could present much more valuable information for comparisons between species.

They do not mention why they established the total number of samples, nor the power of the test.

The conclusion could be a little broader if they mentioned the differences between sampling sites and owner-dog relationships.

Author Response

Reviewer 2

The study is interesting, but they do not show results of the sampling sites between individuals and animals or between individuals, the statistics are poor and from all the samplings that were carried out they could present much more valuable information for comparisons between species.

We are not sure wat the reviewer means with different comparisons between species, as our aim was to determine transmission of strains between humans and dogs, which we showed clearly. It was not the aim to compare the sampling sites. We had included however that we had less strains from different sites in humans than in dogs (Lines 135-139). Other comparisons between the two species (human and dog) were not relevant.

They do not mention why they established the total number of samples, nor the power of the test.

We refer the reviewer to the original description of the isolation of the samples. From this collection of strains described there, we took the S. pseudintermedius strains from dog-human couples that were both positive as to determine whether there was transmission of strains. So the sample size is the number of couples in which we found an S. pseudintermedius isolate in both humans and dogs, we obviously did not include the negative couples. We had no aim to quantify the transmission as for this, a different study design is needed. Multiple couples and multiple samplings per individual (dog or owner) are needed for this as there is a substantial diversity of strains in each individual as we have shown. To determine the transmission frequency, we need first to know the diversity of strains on each individual. Based on that and our results, a sample size and power can be calculated.

The conclusion could be a little broader if they mentioned the differences between sampling sites and owner-dog relationships.

We did not really assess the differences between sampling sites, neither did we assess the owner dog relationship, apart that they were living in the same household.

Reviewer 3 Report

Dear,

The authors have managed to present the comprehensive results of their study on the diversity and zoonotic potential of Staphylococcus pseudintermedius isolated from dogs and their owner. The diversity was investigated using WGS and included MLST, SNPs, detection of virulence-associated and antimicrobial resistance genes, plasmids and phages. This study is valuable not only because of the importance of continuous monitoring and research on S. pseudintermedius, but also because it includes strains isolated from the insufficiently researched region. The research methods are adequately described, and the results are presented in a clear way for the most part (see comments below).

line 77 an increasing interest

line 96 I suggest rephrasing so it's clear that samples were taken from the skin of the abdomen, not abdomen itself.

Figure 1 There is no description for the figure. Please add the label for the first column in the heatmap and explain what those terms represent (e.g. S1S4DN, N156HH1...), or use accession numbers instead. What does the reference 1-448776 represent? If it is a S. pseudintermedius reference strain, why is it blank in the heatmap but it's presented in the dendrogram on the left side of the heatmap? There is no explanation in the material & methods if this was MLST-based or SNP-based phylogeny. Maybe it is stated in the description of the figure which is currently missing, but if not, please add it.

line 114 How did you determine how many SNPs indicates that strains belong to the same clonal group? Please add reference or explain why did choose a specific number of SNPs as a threshold.

line 141 reference - change to numerical

line 150 Is it possible that you did not isolate the plasmids because you did not perform the isolation process specific for plasmids or because of the kit you used?

line 253 may be caused be the fact

line 286 that have been noted before

line 292 Regardless the sampling procedure was explained in more detail in the referenced paper, I suggest you add just 1-2 sentences explaining the procedure in short.

Author Response

Reviewer 3.

The authors have managed to present the comprehensive results of their study on the diversity and zoonotic potential of Staphylococcus pseudintermedius isolated from dogs and their owner. The diversity was investigated using WGS and included MLST, SNPs, detection of virulence-associated and antimicrobial resistance genes, plasmids and phages. This study is valuable not only because of the importance of continuous monitoring and research on S. pseudintermedius, but also because it includes strains isolated from the insufficiently researched region. The research methods are adequately described, and the results are presented in a clear way for the most part (see comments below).

We thank the reviewer for the constructive remarks

line 77 an increasing interest

Adapted line 77

line 96 I suggest rephrasing so it's clear that samples were taken from the skin of the abdomen, not abdomen itself.

Adapted Line 96

Figure 1 There is no description for the figure. Please add the label for the first column in the heatmap and explain what those terms represent (e.g. S1S4DN, N156HH1...), or use accession numbers instead. What does the reference 1-448776 represent? If it is a S. pseudintermedius reference strain, why is it blank in the heatmap but it's presented in the dendrogram on the left side of the heatmap? There is no explanation in the material & methods if this was MLST-based or SNP-based phylogeny. Maybe it is stated in the description of the figure which is currently missing, but if not, please add it.

We aplogise for forgetting the description of the figure, has now been added. We have explained the terms and the reference. The reference strain was S. pseudintermedius reference genome ATCC 49051, it was only used for the SNP phylogeny, not for assessing the accessory genome, therefor it was left empty. We have added to the materials and methods that the phylogeny was based on SNP analysis.

line 114 How did you determine how many SNPs indicates that strains belong to the same clonal group? Please add reference or explain why did choose a specific number of SNPs as a threshold.

Thank you for this remark, this is a good question and still under debate what is now the same strain. We have done it based on what we saw from the data on SNP differences (an excel file has now been added as supplementary material) and what may be accurate in our experience with WGS data. We could see about 3 different groups and this has been stated. But we agree that there is need for a consensus on this.

line 141 reference - change to numerical

Adapted

line 150 Is it possible that you did not isolate the plasmids because you did not perform the isolation process specific for plasmids or because of the kit you used?

We isolated the genomic DNA, and that includes also the plasmids. We do not think we have missed any plasmid replicon (we only determined the plasmid replicons as we did not have closed genomes). Of note, the strains were also not very resistant and as such less plasmids would be expected.

line 253 may be caused be the fact

Adapted now line 261

line 286 that have been noted before

Adapted now line 294

line 292 Regardless the sampling procedure was explained in more detail in the referenced paper, I suggest you add just 1-2 sentences explaining the procedure in short.

We have added some text in the Materials and methods on the overall sampling that was done in the former study.

Reviewer 4 Report

Please consider addressing these points to improve the overall quality and clarity of the manuscript.

1. Firstly, it appears that the sample size utilized in this study is relatively small, which may limit the ability to adequately represent the diverse clonal types of S. pseudintermedius in Trinidad and Tobago, particularly among healthy dogs and their owners.

2. Regarding the Introduction, the authors should provide a more comprehensive background to help readers understand the significance of detecting plasmids. It would be beneficial to elaborate on the relationship between plasmids and antibiotic resistance or virulence. Additionally, the discussion on virulence factors in Staphylococcus pseudintermedius seems limited, as only surface proteins  (spsD/F/P/Q) are mentioned. However, in the Results section, it is noted that the spsJ, spsO, and spsP genes were not detected, while other virulence gene regulators were found in those strains. Expanding upon these aspects would enhance the paper's quality.

3. The use of certain subtitles in the manuscript might be misleading, such as "2.4 Antimicrobial resistance (AMR)" and "2.6. Virulence." This can give the impression that the authors have determined the corresponding phenotypes while they only detected the antimicrobial resistance and virulence genes. To provide clearer descriptions, it is advisable to modify the subtitles to "Antimicrobial resistance (AMR) genes" and "Virulence genes." Furthermore, presenting the results in the form of tables or graphs, rather than relying solely on textual descriptions, would enhance the clarity and directness of the findings.

4. Regarding the statement in lines 240-241, "In this study, only three out of 72 strains were methicillin-resistant and were broadly susceptible to most antibiotics," it would be advantageous to provide supporting evidence or results to substantiate this claim.

Author Response

Reviewer 4

Please consider addressing these points to improve the overall quality and clarity of the manuscript.

  1. Firstly, it appears that the sample size utilized in this study is relatively small, which may limit the ability to adequately represent the diverse clonal types of S. pseudintermediusin Trinidad and Tobago, particularly among healthy dogs and their owners.

The isolates originate from a larger study and only the isolates from dog owner combination were included, as such there is no real sample size in this study but it was in the former study. It was not our aim to determine the clonal diversity in Trinidad and Tobago as this indeed would need a different sampling and sample size estimation. We wanted to determine whether we can detect transfer from dog to human in Trinidad and Tobago.

  1. Regarding the Introduction, the authors should provide a more comprehensive background to help readers understand the significance of detecting plasmids. It would be beneficial to elaborate on the relationship between plasmids and antibiotic resistance or virulence. Additionally, the discussion on virulence factors in Staphylococcus pseudintermediusseems limited, as only surface proteins  (spsD/F/P/Q) are mentioned. However, in the Results section, it is noted that the spsJ, spsO, and spsP genes were not detected, while other virulence gene regulators were found in those strains. Expanding upon these aspects would enhance the paper's quality.

As there were few plasmids, we did not include generalities on plasmids in the introduction.

We only discussed the things that were different.

  1. The use of certain subtitles in the manuscript might be misleading, such as "2.4 Antimicrobial resistance (AMR)" and "2.6. Virulence." This can give the impression that the authors have determined the corresponding phenotypes while they only detected the antimicrobial resistance and virulence genes. To provide clearer descriptions, it is advisable to modify the subtitles to "Antimicrobial resistance (AMR) genes" and "Virulence genes." Furthermore, presenting the results in the form of tables or graphs, rather than relying solely on textual descriptions, would enhance the clarity and directness of the findings.

We have adapted the titles

  1. Regarding the statement in lines 240-241, "In this study, only three out of 72 strains were methicillin-resistant and were broadly susceptible to most antibiotics," it would be advantageous to provide supporting evidence or results to substantiate this claim.

We have adapted as it was missing ‘the other strains’. This is also in the results as well as in figure 1.

Round 2

Reviewer 1 Report

accepted with the revised version.

Reviewer 4 Report

I accept the present form of the manuscript.